# Evolutionary Game Analysis of the Quality of Agricultural Products in Supply Chain

Feixiao Wang and Yaoqun Xu *

Institute of Systems Engineering, Harbin University of Commerce, Harbin 150028, China
* Correspondence: xuyq@hrbcu.edu.cn

**Abstract:** There are many factors affecting the quality and safety of agricultural products in the supply chain of agricultural products. In order to ensure the quality and safety of agricultural products, suppliers and processors need to take their own quality measures to ensure the quality of agricultural products. Quality inspection departments need to strictly supervise suppliers and processors to ensure the implementation of quality measures by both parties. Within the supply chain, the decisions of these three stakeholders are affected by the initial intention, the cost of quality measures, and the penalty amount of the quality inspection department. Outside the supply chain, they are affected by government regulation and consumer feedback. This paper takes the stakeholders in the agricultural product supply chain as the object, brings suppliers, processors, and quality inspection departments into the evolutionary game model, brings the factors that affect the decision-making of these three stakeholders into the model as parameters to analyze the stability of the model in different situations, and then analyzes the factors that affect the decision-making of stakeholders through mathematical simulation according to specific examples. The results show that the enthusiasm of stakeholders to ensure the quality of agricultural products is most affected by the initial intention of each other and the cost of quality measures. At the same time, the punishment of the quality inspection department, the feedback of consumers, and the supervision of the government also play a good role in promoting quality.

**Keywords:** quality and safety of agricultural products; quality measures; agricultural product supply chain; evolutionary game; decisions of stakeholders





## 1. Introduction

The quality of agricultural products in the market has always been a hot topic. With the continuous improvement of people's material living standards, more and more people hope to eat more high-quality and healthy agricultural products [1]. In order to meet the needs of the people, the government and enterprises have been exploring ways to mobilize the stakeholders in the agricultural product supply chain to ensure the quality and safety of agricultural products. In the supply chain of agricultural products, suppliers are mainly responsible for the production of agricultural products and then delivering the produced agricultural products to the processors. The processors process the agricultural products into products that can be eaten by consumers and then sell them to consumers [2]. In the process of the flow of agricultural products, the quality inspection department will conduct quality inspections on the suppliers and processors according to the quality standards [3]. In order to ensure that the quality of agricultural products meets the requirements of the quality inspection department, suppliers and processors will actively take quality measures [4]. However, in real life, in order to ensure the quality of agricultural products, some stakeholders have paid a lot of money to implement quality measures, resulting in their own losses and low returns. As a result of this phenomenon, some stakeholders will seek profits by not implementing quality measures. For example, some agricultural plants will spray large amounts of pesticide during the production of agricultural products [5],

resulting in excessive toxin content. In order to improve the lean meat rate, some meat manufacturers give animals a large amount of clenbuterol. When processing agricultural products, some processors did not strictly screen agricultural products. They added a lot of preservatives to agricultural products and used rent-seeking methods to improve the qualification rate of products [6]. In order to reduce the cost of manpower and material resources, the quality inspection department reduced the inspection intensity, which led to some suppliers and processors being lucky from time to time [3]. As a result, these agricultural products flowing into the market have serious quality and safety risks.

In order to protect people's health and ensure the quality and safety of agricultural products in the market and improve the enthusiasm [7] of stakeholders in the agricultural product supply chain to ensure the quality of agricultural products, the conflict of interest of stakeholders in the agricultural product supply chain must be resolved. On the one hand, the government should not only punish some irresponsible behaviors in the supply chain but also explore the factors that affect the enthusiasm of suppliers and processors to implement quality measures, find out the main reasons why the quality inspection department can carry out strict supervision, and then use policy means to intervene in the decision-making of the stakeholders in the supply chain, so as to improve the enthusiasm of the stakeholders in the supply chain to ensure the quality of agricultural products, and thus maintain the quality and safety of agricultural products [8]. On the other hand, enterprises need to explore the relationship between stakeholders in the supply chain of agricultural products [9], find out the key factors that affect the decision-making of stakeholders, and then provide necessary help in the management and technology of the supply chain. So as to further improve the enthusiasm of stakeholders to ensure the quality of agricultural products [10].

Evolutionary game theory is a theory that combines the analysis of game theory with the analysis of the dynamic evolution process. It emphasizes a dynamic equilibrium. The theory originates from the theory of biological evolution, which has successfully explained some phenomena in the process of biological evolution. Nowadays, economists have made remarkable achievements in analyzing the factors influencing the formation of social habits, norms, institutions, or systems and explaining their formation process by using evolutionary game theory [11–13]. Due to the serious information asymmetry in the flow of agricultural products in the supply chain, coupled with the fact that the quality of agricultural products cannot be easily detected, some stakeholders have the motivation to seek their own interests and damage the interests of others [14], resulting in a certain conflict of interest in the quality of agricultural products among stakeholders in the supply chain [4]. In order to ensure the quality of agricultural products, suppliers must actively take quality measures, such as reducing the use of pesticides, strictly screening mature agricultural products [15], storing agricultural products in cold storage, and transporting them to processors in the form of the cold chain [16,17]. However, some suppliers may reduce their enthusiasm for implementing quality measures and deliver low-quality agricultural products to processors in order to pursue greater benefits. Processors are mainly responsible for the processing of agricultural products. In order to ensure the quality and safety of agricultural products in the supply chain, processors must also actively implement quality measures, such as adopting cold chain logistics to receive agricultural products, preferential processing, rational use of preservatives, use of refrigeration rooms to store agricultural products, ensure the freshness of agricultural products, and finally use cold chain logistics to transport them to enterprises or consumers [18], so as to ensure that consumers can eat fresh, high-quality and healthy agricultural products. However, in order to seek greater benefits, the processors may reduce their enthusiasm for implementing quality measures and may not process agricultural products according to quality standards, causing hidden dangers to consumers. This requires the quality inspection department to be responsible for the quality supervision of agricultural products in the agricultural product supply chain [19]. If the quality inspection department strictly supervises, it will have a strong regulatory effect on suppliers and processors. Once the quality of agricultural

products does not meet the standards, they will be punished accordingly. In addition, outside the supply chain of agricultural products, these three stakeholders are affected by government supervision and consumer feedback [20,21]. Based on the interests of the three stakeholders in the supply chain, and considering the impact of government regulation and consumer feedback, this paper will use the evolutionary game method to establish an evolutionary game model for suppliers, processors, and quality inspection departments in the supply chain of agricultural product, to explore their decisions under the changes of different influencing factors, and provide suggestions for the government and enterprises to better manage the stakeholders in the agricultural product supply chain. To provide countermeasures for better ensuring the quality and safety of agricultural products in the future.

For the quality and safety management of agricultural products, many researchers regard the government, enterprises, consumers, farmers, and agricultural cooperatives as stakeholders, and choose two or three of them as the game players to build a model, change the factors that affect the decision-making of the game players, and let the game players play dynamically and repeatedly until they reach the balance point, so as to draw a conclusion. For example, TENG and CHEN explored the important factors affecting the quality and safety of agricultural products by building an evolutionary game model of government, farmers, and consumers. The results showed that effective government supervision could encourage farmers to produce more green agricultural products, and consumers' efforts to purchase high-quality agricultural products would help improve farmers' enthusiasm for producing high-quality agricultural products [22]. Hong Zhang has explored a new organic agricultural product supply chain by building an evolutionary game model between farmers and the government, and a game model between farmers and customers. The adoption of the organic agricultural product supply chain helps reduce the use of pesticides. The way to improve the adoption rate of the organic agricultural product supply chain is to effectively supervise the government and improve customers' preference for organic agricultural products [23]. Based on new institutional economics and new economic sociology, Liu constructed a three-way evolutionary game model for the production and sales of green agricultural products, analyzed the impact of various factors on the evolutionary game process, and finally found that the proportion and quality contribution of agents, the increase in consumers' preference for green agricultural products, and the increase in social embedded costs are conducive to the effective supply of green agricultural products in the agricultural industrialization consortium [24]. Based on evolutionary game theory, Song and Luo built an evolutionary game model for the government, agricultural enterprises, and telecommunications enterprises, and confirmed that the blockchain technology of telecommunications enterprises has a significant impact on improving the production of green agricultural products [25].

From the previous literature, we can see that many researchers like to take the agricultural product supply chain as a whole object, and study the game relationship between this object and the government, enterprises, and consumers, to explore the influencing factors of these game players on improving the enthusiasm of the agricultural product supply chain to ensure the quality of agricultural products, so as to provide countermeasures for the government and enterprises to manage the supply chain of agricultural product [26], however, these documents ignore the impact of stakeholders' decisions within the supply chain on agricultural products. The quality of agricultural products within the supply chain is mainly affected by suppliers, processors, and quality inspection departments. Although the government has the right to punish the irresponsible behavior of some stakeholders in the agricultural supply chain [27], but if the government wants to manage the quality and safety of agricultural products fundamentally, the government must understand the impact of the decision-making of stakeholders in the supply chain on agricultural products, so as to manage the quality and safety of agricultural products in a policy manner. In order to obtain better benefits, enterprises need to consider not only the feedback influence of consumers on agricultural products, but also the interesting relationship within the

agricultural product supply chain and the factors that affect the motivation of stakeholders to ensure the quality of agricultural products. In this way, enterprises can choose good stakeholders to cooperate [28], better manage the agricultural product supply chain, and provide appropriate help to stakeholders on the factors that affect the quality of agricultural products. Therefore, on the basis of government regulation and consumer feedback, this paper establishes an evolutionary game model for the suppliers, processors, and quality inspection departments in the agricultural product supply chain and conducts evolutionary analysis. Then we use MatLab2018b (Natick, America) for numerical simulation analysis [29]. The factors considered are more comprehensive, and the conclusions drawn are more practical.

## 2. Materials and Methods

Now we use the evolutionary game theory to analyze the game relationship between suppliers, processors, and quality inspection departments in the agricultural product supply chain on the quality and safety of agricultural products in combination with Figure 1, and finally obtain the stability strategy of the evolutionary game model.

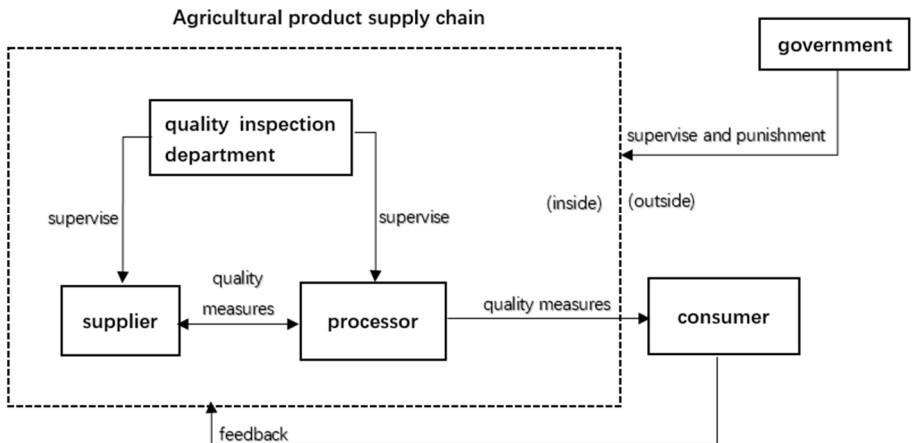

**Figure 1.** Game relationship among suppliers, processors, and quality inspection departments of agricultural products supply chain.

### 2.1. Conditional Assumptions

**Hypothesis 1.** *Suppliers and processors in the agricultural product supply chain have two strategies of actively taking quality measures and negatively taking quality measures on the quality of agricultural products [27]. The quality inspection department has two strategies of strict supervision and regular supervision of the two stakeholders, and the three are characterized by limited rationality and asymmetric information [3,16]. Among them, the probability of agricultural product suppliers actively taking quality measures is $x$ ($0 \leq x \leq 1$), the probability of negative implementation is $(1 - x)$, the probability of agricultural product processors actively taking quality measures is $y$ ($0 \leq y \leq 1$), the probability of negative implementation is $(1 - y)$, the probability of strict supervision by quality inspection departments is $z$ ($0 \leq z \leq 1$), and the probability of regular supervision is $(1 - z)$.*

**Hypothesis 2.** *Agricultural product suppliers actively implement quality measures, and the income obtained is $D_S$. The cost paid is $C_S$. The income obtained by agricultural product processors from actively implementing quality measures is $D_M$. The cost paid is $C_M$. The income obtained from routine supervision of the quality inspection department is $D_T$. The cost of strict supervision is $C_T$.*

**Hypothesis 3.** *when the agricultural product processing industry adopts the "free ride" method and does not implement quality measures in order to obtain greater income [30], the income obtained is $E_M$ ($D_M > E_M > D_M - C_M$), the supplier of agricultural products needs to pay an additional cost*

$\Delta C_S$ on the quality of agricultural products $\Delta C_S (\Delta C_S < C_S)$. At this time, if the quality inspection department strictly supervises, it will impose an appropriate fine on the agricultural product processors. The fine amount is $F_M$. When the agricultural product supplier negatively implements quality measures in the way of "free riding", the income obtained is $E_S (D_S > E_S > D_S - C_S)$, the agricultural product processors need to pay an additional cost $\Delta C_M$ on the quality of agricultural products $\Delta C_M (\Delta C_M < C_M)$. At this time, if the quality inspection department strictly supervises, it will impose appropriate fines on the agricultural product suppliers. The fine amount is $F_S$.

**Hypothesis 4.** *When both agricultural product suppliers and agricultural product processors negatively implement quality measures, there will be no free ride. At this time, consumers will reduce their purchasing power of agricultural products due to the quality and safety problems of agricultural products, and the income of agricultural products will decline due to the quality and safety problems [31]. The loss caused to agricultural product suppliers is $R_S$. The loss caused by agricultural product processors is $R_M$.*

**Hypothesis 5.** *When both agricultural product suppliers and agricultural product processors do not actively implement quality measures and the quality inspection department only carries out routine supervision, serious quality and safety problems of agricultural products will inevitably occur. After putting such agricultural products with serious potential safety risks on the market, they will inevitably be resisted by consumers, and the government will punish the stakeholders in the supply chain [32]. The punishment for agricultural product suppliers is $P_S$. The punishment for agricultural product processors is $P_M$. The punishment for the quality inspection department is $P_T$.*

### 2.2. Model Matrix Construction

Based on the above assumptions, the income matrix of the three-party evolutionary game is constructed as shown in Table 1.

**Table 1.** Strategic income matrix of three stakeholders in agricultural product supply chain.

| Agricultural Products Supplier | Agricultural Products Processor | Quality Inspection Department | |
|---|---|---|---|
| | | **Strict Supervision** | **Routine Supervision** |
| actively implement | actively implement | $D_S - C_S, D_M - C_M, D_T - C_T$ | $D_S - C_S, D_M - C_M, D_T$ |
| | negatively implement | $D_S - C_S - \Delta C_S, E_M - F_M,$ $D_T - C_T + F_M$ | $D_S - C_S - \Delta C_S, E_M, D_T$ |
| negatively implement | actively implement | $E_S - F_S, D_M - C_M - \Delta C_M,$ $D_T - C_T + F_S$ | $E_S, D_M - C_M - \Delta C_M, D_T$ |
| | negatively implement | $E_S - F_S - R_S, E_M - F_M - R_M,$ $D_T - C_T + F_S + F_M$ | $E_S - R_S - P_S, E_M - R_M - P_M, D_T - P_T$ |

### 2.3. Construction of System Dynamic Equation

According to the income matrix of the evolutionary game model, let $U_x$ be the expected income of agricultural product suppliers from actively implementing quality measures, which can be expressed as:

$$U_x = z[y(D_S - C_S) + (1 - y)(D_S - C_S - \Delta C_S)] + (1 + z)[y(D_S - C_S) + (1 - y)(D_S - C_S - \Delta C_S)$$
$$= D_S - C_S - (1 - y)\Delta C_S$$

Set $U_{1-x}$ is the expected income of agricultural product suppliers from the negative implementation of quality measures, which can be expressed as:

$$U_{1-x} = z[y(E_S - F_S) + (1 - y)(E_S - F_S - R_S)] + (1 - z)[yE_S + (1 - y)(E_S - R_S - P_S)]$$
$$= E_S - zF_S - (1 - y)(R_S + P_S) + z(1 - y)P_S$$

According to: $U_x$ and $U_{1-x}$ the average income of building agricultural product suppliers is:

$$\overline{U} = xU_x + (1 - x)U_{1-x}$$

According to the Malthusian equation, the growth rate of agricultural product suppliers' active implementation of quality measures is equal to the income generated from the active implementation of quality measures $U_x$ minus average fitness $\overline{U}$, t is the time, and the copied dynamic equation is sorted out:

$$F(x, y, z) = \frac{\mathrm{d}x}{\mathrm{d}t} = x(U_x - \overline{U}) = x(1 - x)[D_S - C_S - E_S + zF_S - (1 - y)(\Delta C_S - R_S - P_S + zP_S)]$$

In the same way, it can be seen that the reproduction dynamic equation of agricultural product processors actively taking quality measures is:

$$G(x, y, z) = \frac{\mathrm{d}y}{\mathrm{d}t} = y(1 - y)[D_M - C_M - E_M + zF_M - (1 - x)(\Delta C_M - R_M - P_M + zP_M)]$$

The replication dynamic equation of the quality inspection department is:

$$H(x, y, z) = \frac{\mathrm{d}z}{\mathrm{d}t} = z(1 - z)[-C_T + P_T + (1 - x)F_S + (1 - y)F_M - (x + y)P_T + xyP_T]$$

Combining the above replication dynamic equations, it is concluded that the replication dynamic system of agricultural product suppliers, agricultural product processors and quality inspection departments is:

$$F(x, y, z) = x(1 - x)[D_S - C_S - E_S + zF_S - (1 - y)(\Delta C_S - R_S - P_S + zP_S)]$$

$$G(x, y, z) = y(1 - y)[D_M - C_M - E_M + zF_M - (1 - x)(\Delta C_M - R_M - P_M + zP_M)]$$

$$H(x, y, z) = z(1 - z)[-C_T + P_T + (1 - x)F_S + (1 - y)F_M - (x + y)P_T + xyP_T]$$

*2.4. Evolutionary Stability Analysis*

According to the method proposed by Friedman [33], the stability of the evolutionary equilibrium point can be derived from the local stability analysis of the Jacobian matrix of the system. According to the Jacobian matrix (J) constructed by the above replication dynamic system.

$$\begin{pmatrix} \frac{\partial F(x,y,z)}{\partial x} & \frac{\partial F(x,y,z)}{\partial y} & \frac{\partial F(x,y,z)}{\partial z} \\ \frac{\partial G(x,y,z)}{\partial x} & \frac{\partial G(x,y,z)}{\partial y} & \frac{\partial G(x,y,z)}{\partial z} \\ \frac{\partial H(x,y,z)}{\partial x} & \frac{\partial H(x,y,z)}{\partial y} & \frac{\partial H(x,y,z)}{\partial z} \end{pmatrix} = \begin{pmatrix} a_{11} & a_{12} & a_{13} \\ a_{21} & a_{22} & a_{23} \\ a_{31} & a_{32} & a_{33} \end{pmatrix}$$

$$a_{11} = (1 - 2x)[D_S - C_S - E_S + zF_S - (1 - y)(\Delta C_S - R_S - P_S + zP_S)]$$

$$a_{12} = x(1 - x)(\Delta C_S - R_S - P_S + zP_S)$$

$$a_{13} = x(1 - x)[F_S - (1 - y)P_S]$$

$$a_{22} = (1 - 2y)[D_M - C_M - E_M + zF_M - (1 - x)(\Delta C_M - R_M - P_M + zP_M)]$$

$$a_{23} = y(1 - y)[F_M - (1 - x)P_M]$$

$$a_{31} = z(1 - z)[-F_S - (1 - y)P_T]$$

$$a_{32} = z(1 - z)[-F_M - (1 - x)P_T]$$

$$a_{33} = (1 - 2z)[-C_T + P_T + (1 - x)F_S + (1 - y)F_M - (x + y)P_T + xyP_T]$$

Now let $F(x, y, z) = 0, G(x, y, z) = 0, H(x, y, z) = 0$, and the local stable equilibrium point of the system is: $A_1(0, 0, 0), A_2(0, 0, 1), A_3(0, 1, 0), A_4(1, 0, 0), A_5(1, 1, 0), A_6(1, 0, 1), A_7(0, 1, 1), A_8(1, 1, 1)$. According to the theory of evolutionary game, when all eigenvalues

in the matrix are nonpositive, this point can make the replication dynamical system reach the evolutionary stability strategy (ESS). The Jacobian matrix corresponding to each equilibrium point can be obtained by substituting each equilibrium point into the matrix [34]. The eigenvalues of each matrix are shown in Table 2.

**Table 2.** Characteristic values of each equilibrium point.

| Equilibrium Point | $\lambda_1$ | $\lambda_2$ | $\lambda_3$ |
|---|---|---|---|
| $A_1(0,0,0)$ | $D_S - C_S - E_S - \Delta C_S + R_S + P_S$ | $D_M - C_M - E_M - \Delta C_M + R_M + P_M$ | $-C_T + P_T + F_S + F_M$ |
| $A_2(0,0,1)$ | $D_S - C_S - E_S + F_S - \Delta C_S + R_S$ | $D_M - C_M - E_M + F_M - \Delta C_M + R_M$ | $C_T - P_T - F_S - F_M$ |
| $A_3(0,1,0)$ | $D_S - C_S - E_S$ | $-D_M + C_M + E_M + \Delta C_M - R_M - P_M$ | $-C_T + F_S$ |
| $A_4(1,0,0)$ | $-D_S + C_S + E_S + \Delta C_S - R_S - P_S$ | $D_M - C_M - E_M$ | $-C_T + F_M$ |
| $A_5(1,1,0)$ | $-D_S + C_S + E_S$ | $-D_M + C_M + E_M$ | $-C_T$ |
| $A_6(1,0,1)$ | $-D_S + C_S + E_S - F_S + \Delta C_S - R_S$ | $D_M - C_M - E_M + F_M$ | $C_T - F_M$ |
| $A_7(0,1,1)$ | $D_S - C_S - E_S + F_S$ | $-D_M + C_M + E_M - F_M + \Delta C_M - R_M$ | $C_T - F_S$ |
| $A_8(1,1,1)$ | $-D_S + C_S + E_S - F_S$ | $-D_M + C_M + E_M - F_M$ | $C_T$ |

From Table 2, we can see that there are eight possible local stable equilibrium points. To meet the requirements of stable equilibrium points, there must be eight different conditional assumptions.

**Case1:** When $E_S - R_S - P_S > D_S - C_S - \Delta C_S$, $E_M - R_M - P_M > D_M - C_M - \Delta C_M$ and $C_T > P_T + F_S + F_M$, the system tends to the equilibrium stable point (0,0,0). At this time, the income from "free riding" of agricultural product suppliers is greater than the income from "free riding" after deducting the income drop caused by "free riding" and the government's punishment. Therefore, agricultural product suppliers choose not to take quality measures. Similarly, agricultural product processors do not take quality measures. Since the cost paid by the quality inspection department for strict supervision is greater than the sum of the government's punishment and the fine imposed on the agricultural product suppliers and agricultural product processors, it only conducts regular supervision.

**Case2:** When $E_S - R_S - F_S > D_S - C_S - \Delta C_S$, $E_M - R_M - F_M > D_M - C_M - \Delta C_M$ and $C_T < P_T + F_S + F_M$, the system tends to the equilibrium stable point (0,0,1). At this time, the income from "free riding" of agricultural product suppliers is greater than the income from "free riding" after deducting the income drop caused by "free riding" and the punishment by the quality inspection department; therefore, agricultural product suppliers do not take quality measures. Similarly, agricultural product processors do not take quality measures. Since the cost paid by the quality inspection department for strict supervision is less than the sum of the government's punishment and the fine imposed on the agricultural product suppliers and agricultural product processors, the quality inspection department strictly supervises the agricultural product suppliers and agricultural product processors.

**Case3:** When $E_S > D_S - C_S$, $E_M - R_M - P_M < D_M - C_M - \Delta C_M$ and $C_T > F_S$, the system tends to (0,1,0). At this time, the benefit of "free riding" of agricultural product suppliers is greater than the benefit of implementing quality measures. Therefore, agricultural product suppliers do not implement quality measures. The benefit of "free riding" of agricultural product processors is less than the benefit of being "free riding" after the benefit reduction caused by "free riding" and government punishment. Therefore, it is more likely that agricultural product processors will choose to actively implement quality measures. The quality inspection department only carries out routine supervision because the cost of strict supervision is greater than the penalty for agricultural product suppliers.

**Case4:** When $E_S - R_S - F_S < D_S - C_S - \Delta C_S$, $E_M > D_M - C_M$ and $C_T > F_M$, the system tends to (1,0,0). At this time, the income of "free riding" of agricultural product suppliers is less than the income of "free riding" after deducting the income drop caused by "free riding" and punishment by the quality inspection department. Therefore, quality measures will be taken actively. The income of "free riding" of agricultural product processors is greater than the income of quality measures, so no quality measures will be

taken. The cost of strict supervision by the quality inspection department is greater than the penalty amount for agricultural product processors, so regular supervision is required.

**Case5:** When $D_S - C_S > E_S$, $D_M - C_M > E_M$, the system tends to (1,1,0). At this time, the income of agricultural product suppliers from implementing quality measures is greater than the income of "free riding", and the income of agricultural product processors from implementing quality measures is greater than the income of "free riding". Therefore, both parties choose to actively implement the quality measures strategy, and the enthusiasm of quality inspection departments for strict supervision will also be reduced.

**Case6:** When $E_S - R_S - F_S < D_S - C_S - \Delta C_S$, $E_M - F_M > D_M - C_M$ and $C_T < F_M$, the system tends to (1,0,1). At this time, the income from "free riding" of agricultural product suppliers is less than the income from "free riding" after deducting the income drop caused by "free riding" and punishment by the quality inspection department. Therefore, quality measures are implemented. The income from "free riding" of agricultural product processors fined by the quality inspection department is greater than the income from implementing quality measures. Therefore, quality measures are not implemented. The cost of strict supervision by the quality inspection department is less than the fine imposed on agricultural product processors, so strict supervision is carried out.

**Case7:** When $E_S - F_S < D_S - C_S$, $E_M - R_M - F_M < D_M - C_M - \Delta C_M$ and $C_T < F_S$, the system tends to (0,1,1). At this time, the income of "free riding" of agricultural product suppliers after deducting the fines of the quality inspection department is greater than the income of taking quality measures. The income of "free riding" of agricultural product processors is less than the income of being "free riding" after deducting the income drop caused by "free riding" and the fines of the quality inspection department. Therefore, quality measures are taken. The penalty imposed by the quality inspection department on agricultural product suppliers is greater than the cost of strict supervision, so strict supervision is carried out.

**Case8:** When (1,1,1) is substituted into the following equation:

$$(1 - 2z)[-C_T + P_T + (1 - x)F_S + (1 - y)F_M - (x + y)P_T + xyP_T]$$

The result is $C_T > 0$. So (1,1,1) is not an evolutionary stable point. It can be seen that in any case, agricultural product suppliers actively take quality measures, agricultural product processors actively take quality measures, and the strategy of strict supervision by the quality inspection department is difficult to achieve at the same time.

### 3. Result

In order to further explore the influence of the parameters in the model on the strategy selection of the relevant subjects in the agricultural product supply chain, this paper uses MatLab to simulate the dynamic evolution game process of suppliers, processors, and quality inspection departments in the agricultural product supply chain. We adjust the initial intention of the stakeholders, the cost of taking quality measures, the supervision cost of the quality inspection department, the punishment amount of the quality inspection department, the feedback of consumers on the quality and safety of agricultural products, and the punishment intensity of the government. In the simulation process, attention should be paid to the mutual influence of the stakeholders in the agricultural product supply chain, and the stakeholders are all under the same external influence situation. Now, based on the historical data provided by Jiajia limited supply chain Co., Ltd. in Jiamusi Heilongjiang, China, the parameter values are initialized and assumed as follows:

$$D_S = 100, D_M = 80, C_S = 10, C_M = 10, C_T = 20, E_S = 70, E_M = 60, F_S = 20, F_M = 20, \Delta C_S = 5,$$

$$\Delta C_M = 5, R_S = 5, R_M = 5, P_S = 25, P_M = 20, P_T = 15.$$

### 3.1. The Influence of Initial Intention on Evolution Results

From Figure 2, we can see that when the supervision intention of the quality inspection department continues to increase, the speed of system convergence will continue to accelerate, which indicates that the enthusiasm of suppliers and processors to take quality measures will continue to increase. As the enthusiasm of suppliers and processors continues to improve, the enthusiasm of quality inspection departments in supervision will gradually decrease. From the results shown in Figures 3 and 4, we can see that the enthusiasm of suppliers and processors to take quality measures will change with each other. When one party's initial willingness to take quality measures increases gradually, the convergence speed of the system will decrease, which indicates that the increase of one party's enthusiasm will cause the decrease of the other party's enthusiasm.

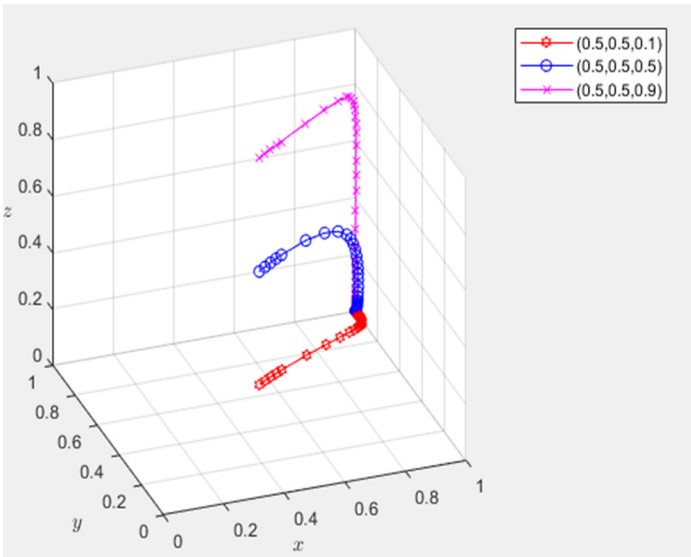

**Figure 2.** Influence of quality inspection department.

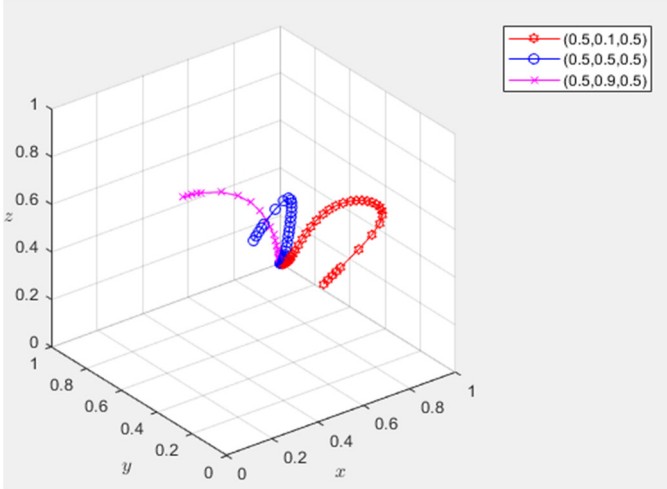

**Figure 3.** Influence of processors.

The results show that if suppliers in the agricultural product supply chain actively take quality measures to ensure the quality and safety of agricultural products, the willingness of processors to take quality measures will be reduced. Similarly, the willingness of suppliers will also decrease with the improvement of the willingness of processors. It can be seen that the wishes of suppliers and processors are mutually restrictive. The improvement of the enthusiasm of the quality inspection department in quality inspection will promote the

suppliers and processors, but it will decrease with the improvement of the enthusiasm of both parties. It can be seen that the quality inspection department will judge whether to strengthen the detection intensity according to the will of the detection object.

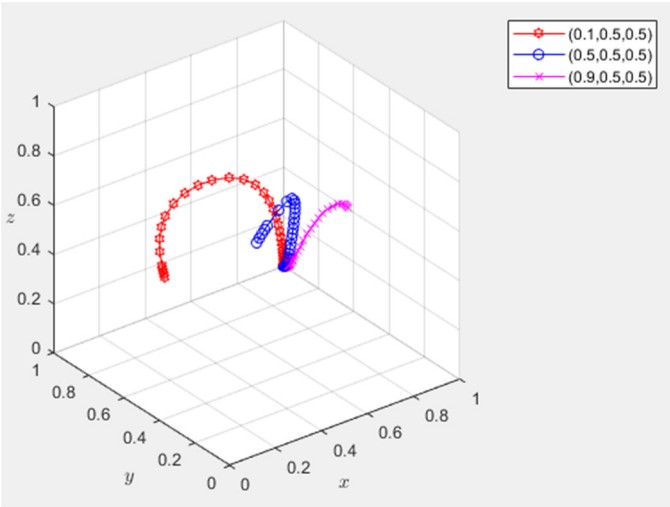

**Figure 4.** Influence of suppliers.

### 3.2. Influence of Cost of Quality Measures on Evolution Results

From Figure 5, we can see that with the continuous increase in the cost of quality measures, the evolution result of the system has changed from the original (1,1,0) to (0,0,1). The cost of quality measures will not only affect the enthusiasm of suppliers and processors to implement quality measures, but also change the supervision enthusiasm of quality inspection departments. When the cost of quality measures becomes 40, the strategy of the processor has changed and no longer takes quality measures. When the cost of quality measures becomes 50, the supplier's strategy also changes, and quality measures are no longer implemented, and the quality inspection department also changes to a strict supervision strategy.

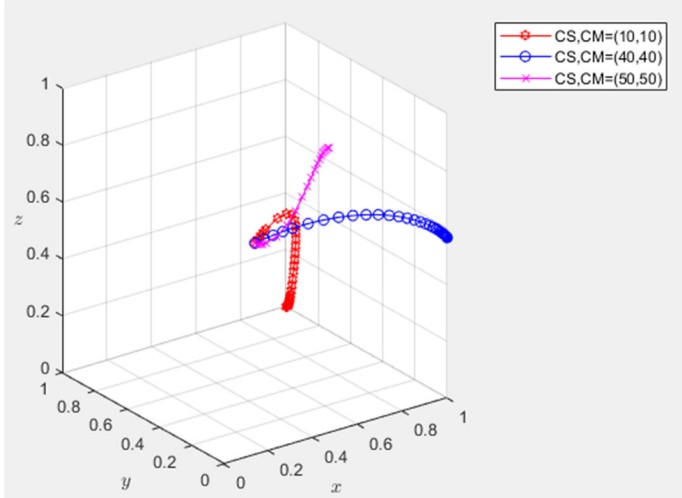

**Figure 5.** Influence of cost of quality measures.

From Figure 6, we can see that when the quality inspection cost of the quality inspection department continues to increase, the system of the quality inspection department will accelerate convergence to regular supervision, which means that the enthusiasm of the quality inspection department for strict supervision will be reduced. At the same time,

the rate at which suppliers and processors converge to take quality measures will also be reduced.

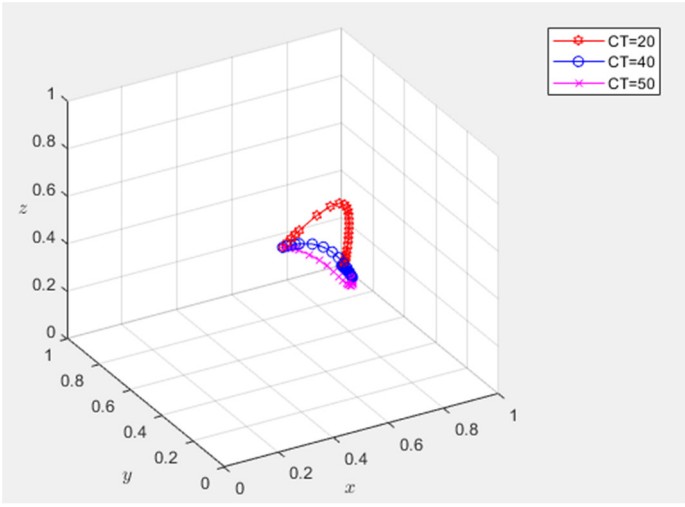

**Figure 6.** Influence of quality inspection cost.

It can be seen from the results that the cost of quality measures has an important impact on the strategic choice of stakeholders in the agricultural product supply chain. The high cost may lead to the stakeholders not taking quality measures and reducing their enthusiasm to ensure the quality and safety of agricultural products, resulting in hidden dangers in the quality and safety of agricultural products.

### 3.3. Influence of Penalty Amount on Evolution Result

From Figure 7, we can see that when the penalty amount continues to increase, the faster the system converges, which means that the increase in the penalty is conducive to the suppliers and processors to actively take quality measures, but the impact of the quality inspection penalty is less than the impact of the cost of quality measures. It shows that although the punishment of the quality inspection department has a certain role in promoting suppliers and processors, it does not substantially change their decisions.

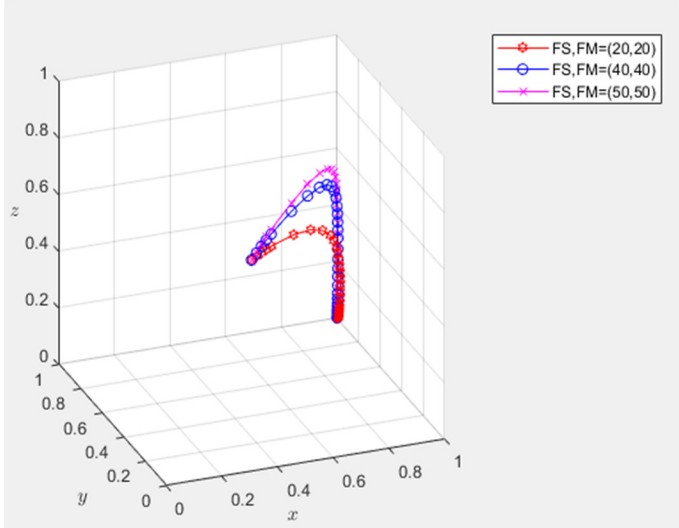

**Figure 7.** Influence of penalty amount.

### 3.4. Consumer Feedback and Government Regulation

Consumer feedback and government regulation are external influencing factors in the agricultural product supply chain, so we need to look at the impact of influencing factors on individual stakeholders separately.

### 3.4.1. Influence of Consumer Feedback

From Figure 8, we can see that when consumers reduce the purchase of agricultural products due to the quality and safety of agricultural products, suppliers and processors in the agricultural product supply chain will be more and more motivated to take quality measures, that is, suppliers and processors will strictly take quality measures to improve the quality of agricultural products for the sake of their own product image and brand image, so as to increase the market share of agricultural products.

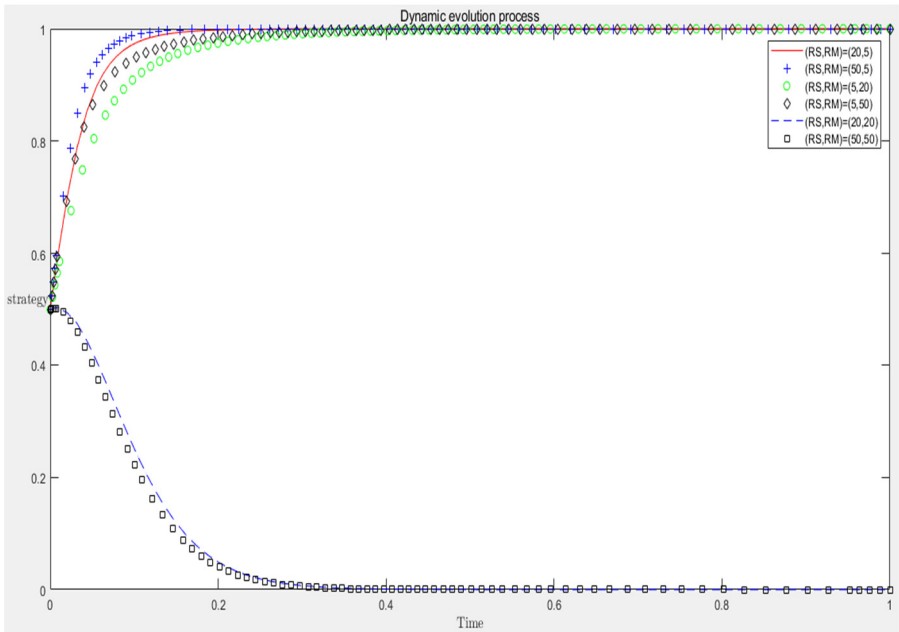

**Figure 8.** Influence of consumer feedback.

### 3.4.2. Influence of Government Punishment

From Figure 9, we can see that when the government increases the punishment of suppliers, processors, and quality inspection departments in the agricultural product supply chain due to the quality problems of agricultural products, the enthusiasm of suppliers and processors to take quality measures will continue to increase, and the enthusiasm of quality inspection departments to carry out strict supervision strategies will also increase, but the overall evolution result of the system will not change.

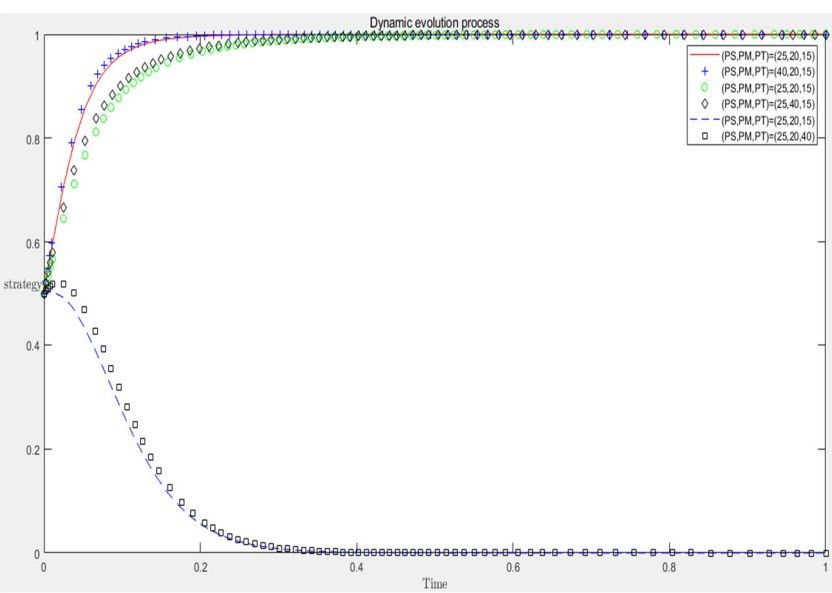

**Figure 9.** Influence of government punishment.

## 4. Conclusions

### 4.1. Conclusion of Evolution Results

From the analysis of the evolutionary game and numerical simulation, it can be seen that the suppliers, processors, and quality inspection departments in the agricultural product supply chain are affected by various factors in ensuring the quality of agricultural products. Among them, the initial intention of the stakeholders and the cost paid to ensure the quality of agricultural products have great influence, which are the fundamental reasons for whether the suppliers and processors take quality measures and whether the quality inspection departments carry out strict supervision. On the one hand, the initial intention of the stakeholders affects and restricts each other in the agricultural product supply chain. When the quality inspection department strictly supervises, the enthusiasm of suppliers and processors to take quality measures will continue to increase. When the willingness of one party of suppliers and processors to take quality measures decreases, the willingness of the other party to take quality measures to ensure the quality of agricultural products will increase. Alibaba's Hema Fresh Food Enterprise knows how to choose a department with good performance to cooperate with itself in the selection and processing of goods so that it can occupy its own position in the market of fresh agricultural products in China [35]. On the other hand, the cost of quality measures plays an important role in ensuring the quality and safety of agricultural products. When the cost of quality measures is low, the enthusiasm of suppliers and processors to implement quality measures will increase. When the cost of quality measures is high, suppliers and processors may not take quality measures, which will also arouse the vigilance of quality inspection departments. When cooperating with the agricultural product supply chain, some e-commerce enterprises will also use financial and technical means to reduce the cost of some stakeholders' investment in the quality of agricultural products [36]. In addition, the punishment amount of the quality inspection department, the quality feedback of consumers on agricultural products, and the government's supervision and punishment will play a good role in promoting suppliers and processors to actively take quality measures and the strict supervision of the quality inspection department.

### 4.2. Suggestions and Prospects

To ensure the quality and safety of agricultural products on the market, the government and enterprises need to make efforts in the following aspects.

When selecting suppliers and processors for cooperation, enterprises should choose merchants with strong brand awareness for cooperation, because such merchants have a strong desire to take quality measures to ensure their own brand image, and the quality and safety of agricultural products supplied and processed by such suppliers and processors are trustworthy. The government should also provide certain welfare policies for these enterprises with strong brand awareness [37] and provide high salaries and high treatment for the staff of the quality inspection department, so as to enhance the enthusiasm of the quality inspection department. In addition, in the agricultural product supply chain, suppliers and processors will lobby and bribe the quality inspection department to bring inferior agricultural products to the market [3]. The government should give appropriate punishment and encourage the masses to report these behaviors so as to further improve the willingness of quality inspection departments to carry out strict supervision.

In terms of quality measure cost, enterprises can provide certain subsidies or technical support for suppliers and processors. For example, cold chain logistics has always been the key point for suppliers and processors to make decisions on quality measures [38]. Enterprises can bear the cold chain logistics costs of suppliers and processors themselves to ensure the freshness of agricultural products in transportation. At the same time, enterprises can provide advanced technology and breeding methods to ensure the quality of agricultural products in a cheaper way. The government can provide some support for the cost of quality measures in terms of electricity and tax, promote the implementation of production, education, and research policies, enable universities to cooperate with enterprises, and develop more efficient methods to ensure the quality of agricultural products [39], so as to reduce the cost required for suppliers and processors to implement quality measures [40]. The supervision cost of the quality inspection department shall also be borne by the government, and the penalty amount of the quality inspection department must be determined according to the performance of suppliers and processors. If the circumstances are serious, they must be severely punished.

For consumer feedback, enterprises should use blockchain technology to establish a traceability system [41] so that consumers can understand which link in the supply chain the quality problem occurs [42]. When consumers become more and more aware of the quality problems of agricultural products, stakeholders in the agricultural product supply chain will actively take measures to ensure the quality and safety of agricultural products.

Finally, as the supervisor, the government should monitor the performance of stakeholders in the agricultural product supply chain in real time and severely punish stakeholders with potential safety hazards to ensure that people can eat high-quality and healthy agricultural products.

*4.3. Research Limitations*

There are many factors that affect the quality of agricultural products inside and outside the supply chain of agricultural products. In order to make the factors considered in the research and analysis more comprehensive, more subjects need to participate in the game and analyze the factors that affect their decision-making. In addition, in order to conduct more in-depth and accurate research, it is necessary to find more data to build more models for analysis so that the results are more consistent with reality and more accurate factors affecting the quality and safety of agricultural products can be obtained.

**Author Contributions:** F.W. is mainly responsible for writing manuscripts, designing and optimizing models, and drawing conclusions through simulation. Y.X. is mainly responsible for the target design and outline design of the manuscript, the writing guidance of the manuscript, and the improvement of the language and grammar. All authors have read and agreed to the published version of the manuscript.

**Funding:** This work was funded by Nature Science Foundation of Heilongjiang, grant number LH2021F035. And this work was supported in part by the Heilongjiang Province philosophy and social science research planning project, grant number 20GLE390.

**Institutional Review Board Statement:** Not applicable.

**Data Availability Statement:** Not applicable.

**Conflicts of Interest:** The authors declare no conflict of interest.

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
