# Peer review of "Evolutionary Game Analysis of the Quality of Agricultural Products in Supply Chain"

_agriculture, doi:10.3390/agriculture12101575_

Round 1

Reviewer 1 Report

[No line numbers were given in the text. For this reason, comments will be linked to the chapter number and manuscript page.]

The study is interesting. An important issue of food quality in the supply chain was addressed. The method used is suitable for simulating the behavior of system participants and has been well used. I only suggest minor additions or corrections.

In the abstract, it should be emphasized that the presented results are based on certain assumptions about the relationship between income, costs, penalties, the possibility of consumers opting out of the purchase. This means that results may differ if assumptions changed. A few suggestions from the summary, e.g. about the importance of technological progress, can also be added to the abstract.

[page 2, ch. 2] In the literature review, little attention is paid to the different areas in which EGA is used. It is worth extending the literature review. Many items (e.g. 10.3934 / mbe.2021320, 10.1016 / j.resconrec.2020.104863, 10.3390 / ijerph16152775, 10.1080 / 0952813X.2010.506300) present not only the directions of application, but also take into account the role of penalties and subsidies. There are also indicated restrictions on the use of the results of this method. The assumptions and results obtained in the model give only a demonstration, which may differ from reality, but indicate the directions of action challenging to find based on observation of reality.

It is worth compiling the parameters resulting from the assumptions in one table and providing their description and range of values there. (page 4).

On page 5, retailers are mentioned. Is it about processors?

Part 6 should also briefly outline the test limitations. Especially if they result from assumptions or simplification of reality.

There are typos in the text, spaces are often missing. In addition, some equations (e.g. on page 5) may be written in one line.

Author Response

Thank you very much for your suggestions on this manuscript. We have made major changes to this manuscript. The details are attached.

Reviewer 2 Report

The actors presented an article with an unusual structure, but it is consistent and understandable for the recipient.

The theory of cooperative games in making logistic decisions indicates that cooperation allows you to generate benefits and synergistic effects and can be successfully applied in the decision-making process in the entire supply chain. This is not a new issue, but in this case it is presented in an interesting way.

The article is coherent and thoughtful, it indicates the importance of using cooperative games in the logistics of the supply chain. The text is well written, coherent and logical. The abstract captures the meaning of the work well. The goals and hypotheses coincide with the conclusions. Correct inference.

However, in the first part, the introduction suggests using quotes from other authors to verify the content or combining chapters 1 and 2 and completing the citation. I would replace the terms index with keywords. The article also lacks a juxtaposition of theories presented by the authors with the results of research by other authors.

In conclusion, I highly rate the article, but I think that it needs to be placed among the research of other authors. Moreover, is the research methodology proprietary or based on previously known theories? Please complete this information. Finally, it suggests enlarging the graphs and creating equations in the editor.

Author Response

Thank you for your comments on this manuscript. We have revised the content of the manuscript according to your suggestions and sent it to the editorial department. I believe it will be in your hands soon.

Reviewer 3 Report

1. Citations of good articles are needed. The introduction section has no citations.

2. The novelty of this paper is the method. However, the author(s) failed to explain why we need to look into this method. The author (s) did not provide problems that require the method to be employed. Thus, the author(s) should start with the problem in the introduction section.

3. Author(s)  mentioned some fundamental problems, such as chemicals not following standards and processors not being stringent. Findings in the literature should back these problems. 

4. "Previous studies focused on the interesting relationship between agricultural producers and consumers, the relationship between the government and agricultural producers, and the relationship between the government and consumers. However, it ignores the mutual restriction between stakeholders in the supply chain of agricultural products (the influence of mutual decision-making among stakeholders)". There must be a reason for the past literature not doing this. The author (s) should cite the articles and see why they did not cover the area. By doing so, the author(s) can strengthen the paper's originality.

5. Author(s) jumped to the literature review section without explaining the problem worth solving other than previous literature did not cover the area of research. Author(s) should also include research objectives so that it is easier to understand what author(s) want to investigate and what method is suitable to answer the objective.

6. Literature review section does not contain a critical literature discussion. What has been found in the literature? What variables are involved in this paper? What is still lacking, and what will the author (s) do differently to fill the gap?

7. Section 3 under section A should be accompanied by a diagram explaining the process of the agriculture supply chain so that it is easier to understand why game theory is needed.

8. Figure 1 suddenly has a government, supplier, process, and consumer. Thus, I recommend that the author(s) put in the agriculture supply chain process, and the explanation should be covered in section 2, literature review. 

9. The hypotheses have no citations. How do we validate the assumptions are correct? For example, "The quality inspection department has two strategies of strict supervision and regular supervision on the two stakeholders, and the three are characterized by limited rationality and asymmetric information". How do we validate the two strategies? Where did this information gather? The rest also has similar issues.

10. Author(s) mentioned the method proposed by Friedman. No citation and no explanation of what this method is about. No discussion on previous studies either.

11. Numerical simulation should have more explanation on its closeness with a real-world example. The author (s) can cite similar cases so that the context can help to explain and help other researchers to replicate the study.

12. The findings focus more on quality. Other than quality, there are other operational performance measures that stakeholders would like to see, such as flexibility, responsiveness, environmentally friendly and so on. The author (s) should use the finding to explain the overall performance but be specific when recommending practical improvements.

13. The findings are not new and already found in the literature. The method is interesting, but again, without a specific problem, we could not see this paper's originality.

Author Response

(The authors gave the same response as above.)

Reviewer 4 Report

I would like to reject this paper because of several reasons. First, the very low readability across the whole study. If the author(s) can do a thoroughly proofreading, I would like reconsider this study. Second, introduction is inappropriate. I do not know what they want to do of this work. Third, literature review is inappropriate. Without proper reviewing of other articles in this area, it is difficult to argue contributions. 

Author Response

(The authors gave the same response as above.)

Round 2

Reviewer 2 Report

The authors have significantly improved the manuscript. I recommend it for publication in its present form. 

Author Response

Dear Reviewer,

Thank you for your approval of this manuscript. We have repeatedly reviewed this manuscript and made some modifications to make it better. Thank you for your guidance and suggestions on this manuscript.

Kind regards,
Feixiao Wang
First author

Reviewer 3 Report

References are from Sustainability. However, other than Sustainability and Food Control, not many high quality journals are cited. Please find quality journals of Q1 Q2 in WoS.

Author Response

Dear reviewer,

Thank you for your approval of this manuscript. We have repeatedly reviewed this manuscript and made some modifications to make it better. Thank you for your guidance and suggestions on this manuscript.

Kind regards,
Feixiao Wang
First author

Reviewer 4 Report

The author(s) successfully tackled all the comments that I raised in the last version. Thus, I suggest to accept this paper in its current form. 

Author Response

Dear reviewer,

Thank you for your approval of this manuscript. We have repeatedly reviewed this manuscript and made some modifications to make it better. Thank you for your guidance and suggestions on this manuscript.